# Human rDNA and Cancer

**DOI:** 10.3390/cells10123452

**Published:** 2021-12-08

**Authors:** Evgeny Smirnov, Nikola Chmúrčiaková, Dušan Cmarko

**Affiliations:** Laboratory of Cell Biology, Institute of Biology and Medical Genetics, First Faculty of Medicine, Charles University and General University Hospital in Prague, 128 01 Prague, Czech Republic; nikola.chmurciakova@lf1.cuni.cz (N.C.); dusan.cmarko@lf1.cuni.cz (D.C.)

**Keywords:** human rDNA, ribosomal genes, IGS, cancer, non-coding RNA, copy number

## Abstract

In human cells, each rDNA unit consists of the ~13 kb long ribosomal part and ~30 kb long intergenic spacer (IGS). The ribosomal part, transcribed by RNA polymerase I (pol I), includes genes coding for 18S, 5.8S, and 28S RNAs of the ribosomal particles, as well as their four transcribed spacers. Being highly repetitive, intensively transcribed, and abundantly methylated, rDNA is a very fragile site of the genome, with high risk of instability leading to cancer. Multiple small mutations, considerable expansion or contraction of the rDNA locus, and abnormally enhanced pol I transcription are usual symptoms of transformation. Recently it was found that both IGS and the ribosomal part of the locus contain many functional/potentially functional regions producing non-coding RNAs, which participate in the pol I activity regulation, stress reactions, and development of the malignant phenotype. Thus, there are solid reasons to believe that rDNA locus plays crucial role in carcinogenesis. In this review we discuss the data concerning the human rDNA and its closely associated factors as both targets and drivers of the pathways essential for carcinogenesis. We also examine whether variability in the structure of the locus may be blamed for the malignant transformation. Additionally, we consider the prospects of therapy focused on the activity of rDNA.

## 1. Introduction

In human cells, ribosomal DNA (rDNA) is arranged in ten clusters of multiple tandem units situated on the short arms of five chromosomes (#13, #14, #15, #21, and #22) [1]. Each rDNA unit typically consists of the ~13 kb-long ribosomal part, a single transcription unit coding for 45S (or 47S) primary rRNA, and the ~30 kb-long intergenic spacer (IGS). The ribosomal part is transcribed by RNA polymerase I (pol I) and includes three genes coding, respectively, for the 18S, 5.8S, and 28S rRNAs of the ribosomal particles. The genes are separated by internal transcribed spacers, ITS1 and ITS2, and flanked by external transcribed spacers, 5′ ETS and 3′ ETS [2]. IGS has a more complex structure. It contains promoter, multiple repetitive sequences of different length, as well as various functional, or supposedly functional, regions producing non-coding RNAs which participate in the pol I activity regulation, stress reactions, and other processes in the cell [3]. In the metabolically active cells, which require permanent supply of new ribosomes, expression of the ribosomal genes is very intensive throughout interphase. At the end of mitosis, the pol I begins to transcribe ribosomal genes, making primary nucleoli, which are subsequently joined by the rRNA processing factors and the factors responsible for the assembly of ribosomes. Thus, the definitive nucleoli emerge as dynamic structures based on the rDNA [4,5]. The ribosomal function (i.e., production of rRNAs and assembly of the ribosomal particles), is the main function of nucleoli, in which rDNA takes a great part. But recent studies have revealed numerous extra-ribosomal nucleolar functions, such as regulation of cell cycle progression, senescence, assembly of signal recognition particles, modification of transfer RNAs and response to stress [6,7,8]. The central organizing role of rDNA in the nucleolus suggests that ribosomal genes and/or IGS sequences play a significant part in any of these functions.

In this review we focus on the data concerning the human rDNA and its closely associated factors as both target and regulator of the pathways essential for carcinogenesis. We also examine whether some variability in the structure of the locus, may trigger the malignant transformation. Additionally, we consider the therapy targeting the activity of rDNA.

## 2. Nucleoli and Cancer

Nucleolus is a highly dynamic structure and quickly responds to changing proliferative or environmental cues, including alterations of the major tumor suppressor pathways, which makes this organelle a mirror of the malignant transformation [7,9]. Morphologically aberrant nucleoli revealed by the simple method of silver staining appear in various kinds of cancers, and usually reflect the defects in rDNA regulation [10,11]. Since entry of a cell in the proliferative cycle is always associated with increased production of ribosomal RNA (rRNA) and ribosomes, it is no wonder that cancer cells typically have increased, sometimes gigantic, or multiplied nucleoli [9]. The evaluation of nucleolar size may provide reliable information about the tumor. Thus, in a series of studies carried out on numerous human tumors, nucleolar hypertrophy was shown to be an independent prognostic parameter of a fatal clinical outcome, and high rates of rDNA transcription correlated with adverse prognosis [9]. Measurement of nucleolar size have also been used to assess the response to chemotherapeutic drugs [12,13,14]. However, importance of nucleolar changes in tumor pathology is still not sufficiently understood [7]. Apparently, through their ribosomal function nucleoli become an essential target of the carcinogenic signaling. Hence, overexpression of oncogene proteins or loss of tumor suppressor activity result in the consistent hyperactivation of rDNA transcription, which leads to dysregulation of both ribosome biogenesis and extra-ribosomal nucleolar functions [7]. The nucleoli are regarded as upstream regulators of the cell transformation or adaptors to the cancer-associated stress. Importantly, activation of the tumor suppressor function is mediated, in part, by the nucleolus. This may be affected, for example, through coordinated transport of p53 and ribosomal subunits [7,15]. The nucleolus also seems to play an active role in the DNA damage response network. For instance, in the cells subjected to ionizing radiation, double stranded break (DSB) repair factors RNF8 and BRCA1 translocate from the nucleolus to DNA-damage response foci in the nucleoplasm, and come back to the nucleolus several hours later, presumably following DSB repair [16]. It has been supposed that nucleoli are highly sensitive detectors of DNA damage in the cell nucleus and serves as a protector of the entire genome by activating p53-dependent or independent DNA damage repair [17,18].

Therefore, disorganization of either ribosomal or extra-ribosomal functions of nucleoli may be a key permissive step for malignant transformation [7].

## 3. Proteins Associated with rDNA and Cancer

### 3.1. Pol I Complex

Tumors are characterized by high levels of ribosome production due to their reliance on cell proliferation and increased protein synthesis. Therefore, transcription of rDNA by pol I, which is a crucial step in ribosome biogenesis, plays a central role in cancer [6,19,20,21,22,23]. Multiple signaling pathways engaged in the malignant transformation converge directly upon the pol I machinery [7]. Thus, tumor suppressor factors interact with the upstream binding factor (UBF) and selectivity factor-1 (SL-1), the key components of the pol I pre-initiation complex [24]. The CDK4-cyclin D and CDK2-cyclin E/A, which ensure entering S phase, can enhance pol I transcription rates through activating phosphorylation of UBF [25]. Additionally, the abnormal increase of pol I activity can corrupt extra-ribosomal functions of the nucleoli, and such corruption seems to be a critical determinant of malignant transformation [7]. It has been suggested that accelerated ribosome biogenesis is a key factor in the development of malignant phenotype [26]. Remarkably, most of the currently used chemotherapy drugs inhibit rDNA transcription and/or other steps of the ribosome biogenesis [27], and some new preparations are designed to target pol I specifically [7,26].

### 3.2. Myc

The transcription factor Myc (product of the c-Myc oncogene) is one of the most frequently activated oncoproteins, overexpressed in ~50% of all cancers [28]. Myc is a key transcription factor for proliferation that can also affect DNA replication [29]. This protein, which regulates transcription of many genes, particularly those driving cell growth, is accumulated in the nucleoli and is closely associated with rDNA, where it has several binding sites in the promoter and terminator area. In the active ribosomal genes Myc, together with transcription termination factor-1 (TTF-1), mediates formation of loops, in which the promoter is joined to the terminator. These loops seem to be necessary for maintaining the nucleolar chromatin structure favourable for transcription [30]. Myc also modulates the rRNA synthesis by upregulating expression of the pol I subunits and transcription factors, such as UBF and RRN3, and by associating with SL-1 to stabilize the UBF/SL1 complex [29,31,32,33]. In such a way, one of the most important oncoproteins is closely associated with the ribosomal function of rDNA.

### 3.3. Tumor Suppressors p53 and Rb

The two major tumor suppressor proteins, p53 and retinoblastoma protein (pRb), control proliferation in mammalian cells. In human cancers, the function of the suppressors is frequently altered, so that they may be involved in tumorigenesis. These changes lead not only to the loss of control on cell proliferation, but also to up-regulation of rRNA transcriptional activity [9]. Protein p53 can trigger apoptosis or cell cycle arrest at G2 phase in response to cellular stress. Mutated forms of this protein appear in approximately half of all human tumors, and even when the wild type is still preserved, its function is compromised in most cases [34]. Pol I transcription may be repressed as a result of p53 binding to its consensus site in IGS [35]. Protein p53 also inhibits Pol I transcription by association with SL-1, thus preventing its interaction with UBF and formation of the pre-initiation complex [36]. Additionally, the downstream targets of p53-dependent stress signaling negatively regulate pol I transcription, including activation of pRb, or repression of Myc [37,38]. On the other hand, inhibition of rDNA transcription results in a p53-mediated cell cycle arrest [6]. The tumor suppressor pRb prevents entering cells in S phase of the cell cycle. Loss of pRb activity (e.g., as a result of cyclin D overexpression), CDK4 and CDK2 hyper-activation, or activation of caspase dependent proteolytic pathways, is observed in many cancers [39]. Normally pRb inhibits rDNA transcription by interacting with UBF, which prevents its interaction with SL-1 in the pre-initiation complex, and/or by binding directly to the promoter [24].

### 3.4. Basonuclin

Basonuclin is a zinc-finger protein usually found in epithelial and germ cells, but its expression is very high in basal cell carcinoma of skin. Normally this protein binds to rDNA promoter and takes part in the initiation of the pol I transcription. Apparently, dysregulation of basonuclin function may contribute to carcinogenesis [40].

## 4. Methylation of rDNA and Cancer

DNA methylation is one of the major factors which determine the gene expression and progression of human diseases is often accompanied by changing the rDNA methylation pattern [14,41,42]. The locus of human rDNA is distinguished by high levels of DNA methylation. Typically, one unit contains >1500 CpG islands (i.e., >10 CpGs per 100 nt) [43,44]. Studies have shown that cancer correlates with altered methylation status of the rDNA locus [42]. Some of these alterations may be regarded as more or less efficient self-defence of the cell against transformation. Thus, in ovarian cancer patients who have long progression-free survival the level of rDNA methylation was higher than in norm [45]. Interestingly, a similar phenomenon is observed in aging and this aspect of senescence appears to be an adaptation to the increased risk of malignant transformation [43,46,47]. Indeed, signaling pathways of senescence involve oncogenes and tumor suppressors, which respectively promote and inhibit the uncontrolled growth and proliferation in cancer. It is clear that the processes of DNA methylation, aging, and malignant transformation are closely knitted together [7].

The distribution of the CH3 groups along the DNA strand presents a special problem. The methylation of promoter is usually believed to be a repressive epigenetic mark, although in senescent cells the rRNA coding regions are highly methylated as well [42,48]. Strangely enough, methylation of the ribosomal genes positively correlates with their transcription intensity, which is known as “the gene-body paradox“ [49]. Besides, both hypo- and hypermethylation may be found in the abnormally functioning rDNA locus. Thus in the late-stage ovarian cancer, breast cancer, primary endometrial carcinoma, liver cancer, cervical cancer (cervical intraepithelial neoplasia), and hepatocellular carcinomas, the hypomethylation was found in the promoter [41,45,50,51,52,53] but the hypermethylation appeared frequently in some transcribed spacer regions, in IGS, and in the regions coding for 28S and 5.8S rRNA [42]. On the other hand, in patients with primary oral squamous cell carcinoma and prostate cancer, the locus seemed to be methylated as in norm [14,54]. But in the latter case some subtle changes in pattern of methylation might have eluded detection. It should be also kept in mind that rise of the pol I activity, which is often observed in the transformed cells, does not necessarily correlate with hypomethylation of the promoter, since rDNA transcription may be intensified or inhibited in many alternative ways [14,55].

Thus, methylation of rDNA locus, especially its promoter, is one of the key factors of carcinogenesis, but the phenomenon is complicated.

## 5. Causes of rDNA Instability

Being highly repetitive, heavily transcribed, and methylated, rDNA is a very fragile spot of the human genome, with high risk of instability leading to cancer [8,56] (Figure 1). Repetitive sequences are characterized by frequent recombination, and when the recombination (e.g., occurring after double strand breaks (DSBs)) is dysregulated, it becomes a potential driving force of genomic instability and cancer phenotype [57,58,59,60]. There are multiple hot-spots of DSB in human rDNA; the most prominent of them identified within IGS were named ‘’Pleiades”. These Pleiades being nine in number (unlike the celestial ones) contain binding sites for the multifunctional CTCF protein and active chromatin H3K4me3 mark [58,61].

Repair of the damaged rDNA repeats may engage sister chromatids, homologous as well as non-homologues chromosomes, or even repeats in cis position as repair templates. However, the restoration of damaged sites may be accompanied by interchromatid crossovers, translocations, generation of acentric/dicentric chromosomes, repeat expansions and contractions, and/or extra-chromosomal circles [59,62]. Thus, 28S rRNA has been identified as a novel fusion partner (produced by translocation) of carcinoma-related genes such as BCL6, BCL11B, IGKV3-20, and COG1 in gastric lymphoma or hematopoietic tumors [63,64].

Normally, the unfavorable changes in the locus are suppressed by the packaging of repetitive sequences as constitutive heterochromatin, which is typically associated with histone H3 lysine 9 methylation, the heterochromatin protein 1a (HP1a) binding, histone hypoacetylation, and expansive DNA methylation. Such chromatin structure potentiates pairing, regulates repair, and inhibits recombination [59,65]. Therefore, mutations occurring, for instance, in the genes of methyltransferases, as well as environmental factors that compromise heterochromatin function, may be responsible for epigenetic instability leading to cancer and other diseases [56,66].

Small-scale DNA polymorphism represented by single nucleotide variants (SNV), short insertions and deletions, which regularly occur as errors of replication, seems to be another important factor of carcinogenesis. Correlation between malignant transformation and high frequency of the small-scale variations have been well demonstrated [67,68]. In epithelial cancer cells, a significant increase in SNV was found in the region between −388 bp and +306 bp, which includes sequences coding for pRNA and PAPAS involved in the inhibition of the rRNA synthesis [67].

But, although rDNA instability seems to be a common feature of cancer and other severe health problems [23,69], it is still not clear, whether any alterations of the rDNA structure or function are able to initiate carcinogenesis. Furthermore, the potential causes of transformation may be sought among both the single and multiple mutations, depending on the kind of tumor. In adult patients with lung and colorectal cancer, over half of the solid tumors showed multiple changes in the rDNA sequence as compared to the surrounding non-tumor tissue or normal peripheral blood. At the same time, in the pediatric patients suffering from leukemia the frequency of changes was greatly reduced. This seems to agree with the idea that pediatric leukemia is driven by specific dangerous mutations, rather than by numerous low-impact negative variants accumulated gradually in the course of human generations [21].

Probably the best studied diseases connected with disruption of the ribosome biogenesis are ribosomopathies. These rare diseases increase the risk of carcinogenesis, but they all seem to be consequences of mutation in ribosome protein and ribosome-related processor genes, not in the rDNA locus itself [70,71].

It has been demonstrated that malignant transformation correlates with instability of rDNA locus; but so far this instability has not been proved to be a primary cause of carcinogenesis.

## 6. Variability of rDNA Copy Number in Cancer

Even in normal state, the number of copies in the rDNA locus varies among individuals, individual cells, and the rDNA bearing chromosomes within one cell. But this variability increases usually by 20–80% in pathology, especially when the cells respond to the DNA damage [59,72,73,74]. One may reasonably expect that changes in rDNA copy number would lead to changes in metabolism, differentiation program, chromosome stability, epigenetic state, particularly affecting protein synthesis rates, quality control, and protein homeostasis [7,21]. Considerable expansion or contraction of the rDNA locus often follow unequal sister chromatid exchanges and may be regarded as symptoms of malignant transformation [72,75,76,77]. Analysis of a subset of human lung and colon cancers revealed repeat expansion in one-third of the tumors [21]. Amplification of the locus may be connected to the increased production of ribosomes in the transformed cells. But generally, in tumors as well as during aging, the copy number of rDNA may decrease as well as increase [42,59,76,78,79]. It was proposed that rDNA copy numbers become unstable when the stabilizing function of heterochromatin is compromised [59]. Some authors believe that abundance of copies itself increases stability of the locus by providing normal sequences for DSB repair, and that cancer cells with low copy number may be more sensitive to DNA damaging agents [15,73]. But other authors argue that hyper-variability of the locus size is rather a general outcome of the malignant transformation [59]. According to many studies, both losses and gains of rDNA repeats may correlate with predisposition to cancer, premature aging, and neurological impairment in ataxia-telangiectasia and Bloom syndrome [21,59,62,72,76,80,81].

The increased instability of rDNA copy number may have multiple effects on the nucleolar and nuclear functions including telomere maintenance, transposable element silencing, and satellite DNA stability, which are important factors of tumor suppression [59,82,83]. But since quantification and monitoring of the rDNA copy numbers are hindered by the great heterogeneity of tumors, the role of the rDNA locus size variability in carcinogenesis has not been sufficiently studied [21,83].

## 7. Functional Regions of rDNA and Cancer

The distribution of the functional regions in the locus are shown in Figure 2. Production of rRNA, which is the main function of rDNA, was treated in Section 2 and Section 3. Here we focus on other functions of the locus.

### 7.1. Micro-RNA (miRNA) Produced by rDNA

The RNAs of 21–23 nt long and various structure, involved in gene expression and other processes, are known as miRNAs. It is believed that all human genes may fall under control of these short molecules [84]. Even small changes in the miRNA profile can alter such functions as immunoreactivity, cell proliferation and differentiation, apoptosis, and tumor suppression. Most miRNA coding sequences are transcribed by RNA polymerase II as hairpin-shaped primary miRNA which is processed into pre-miRNA after cleavage of the 5′-cap and 3′-polyA tail by the microprocessor complex. Typically, pre-miRNA is exported to the cytoplasm by exportin-5 and further processed into the miRNA duplex, which is bound to Argonaute protein (AGO) and loaded into the RNA-induced silencing complex (RISC) [84]. This complex, recognizing complementary sequences in its target mRNAs, can induce transcript destabilization (low complementarity) or transcript cleavage (high complementarity) [85,86,87,88]. It has been supposed that transcription of DNA in response to a stress cannot produce enough miRNA molecules for efficient defense reaction of the cell, and some of these molecules must be derived from ready-made RNAs, such as rRNA and tRNA. Multiple miRNAs of various origin have been detected in nucleolus by in situ hybridization [89]. On the other hand, many miRNAs operating both within and outside the nucleoli correspond to the rDNA sequences [85]. Some of these ribosomal miRNAs (rmiRNAs) are abundantly expressed [90,91]. But despite their relatively high levels and potential significance, rmiRNAs have been often removed from the RNA-sequence analyses as supposed by-products of rRNA degradation [86]. Now sequences coding for rmiRNAs are found both in IGS and the ribosomal part of the rDNA locus, but they are particularly concentrated in the three ribosomal genes [85,92]. Interestingly, according to the existing datasets, many rmiRNA coding regions match sequences in other parts of genome, but some miRNAs can only be attributed to rDNA [85,91,93]. It seems that rmiRNAs are produced by pol I, and then processed in a manner different from other miRNAs [91], although according to some reports, expression of miRNAs was resistant to pol I inhibition [94]. Human rmiRNAs participate in numerous ribosomal and extra-ribosomal functions of the nucleoli, including regulation of the rDNA copy number and expression of the Alu sequences [91]. Analysis of rmiRNA producing sequences using the Kyoto Encyclopedia of Genes and Genomes (KEGG) indicated that the majority of the corresponding RNAs are involved in cancer or cancer-related pathways such as MAPK signaling and ERBB signaling. This suggests that rmiRNAs are inclined to target cancer-related genes and might have some important roles in anti-cancer or anti-stress pathways [91]. Earlier it has been known that rmiRNAs such as miR-663, miR-1275, miR-3656, miR-3687, and miR-4417 are associated with tumor suppression, carcinomas, and breast cancer [95,96,97]. Ribosomal miRNA from human ETS1 induces endothelial inflammation and atherosclerosis [98]. Ribosomal miRNAs interact with ribosome production and their dysregulation may be implicated in ribosomopathies at the level of transcription, processing, and post-translational events [88]. Some rmiRNAs are involved in two classical hematological ribosomopathies, 5q-myelodysplastic syndrome and Diamond-Blackfan anemia [88,99,100]. Importantly, miRNAs produced by various rDNA regions may serve in cancer diagnosis [91].

### 7.2. Small rRNAs (srRNA)

This group of miRNA-like molecules, varying in size and structure, derive from the precursor and mature rRNAs, as well as IGS, in a poorly understood manner. Some srRNAs may emerge as mere fragments of random degradation, which is a necessary phase of rRNA metabolism [86]. But many of these small molecules seem to be stable products of a controlled processing which may be Drosha-dependent or independent. Unlike miRNAs, they are not always associated with AGO complexes, as established by analysis of the available immunoprecipitation and high-throughput sequencing data. Studies also reveal that srRNAs are involved in regulation of ribosome biosynthesis and cancer related pathways [85]. Several classes of small RNA are described, and all of them seem to be related to carcinogenesis. Moreover, a lot of miRNA-like molecules in the human cells probably elude detection or need further investigation [88]. At least four classes of the small RNAs may be listed among srRNA.

### 7.3. QDE-2-Interacting Small RNAs, or qiRNAs

QDE-2-interacting small RNAs, or qiRNAs are interference RNAs induced by DNA damage. QiRNAs originate mostly from the rDNA locus and participate in DNA damage response in the filamentous fungus Neurospora crassa. It has been reported, that in human cells qiRNA may be affected in diabetes, Perlman syndrome, Wilm’s tumor development, early embryogenesis, and stem cell proliferation [101,102].

### 7.4. Guide RNAs (gRNA)

gRNA typically binds to DNA and anti-sense RNA sequences by complementary base pairing. Binding of gRNA to RNAs results in their post-transcriptional modifications, such as ribomethylation and introducing pseudouridine. Thus, modified RNA and DNA sequences may be targets of deletion, insertion and other alterations. Some of human gRNAs originate from rDNA locus [85].

### 7.5. Piwi-Interacting RNA (piRNA)

*Piwi-interacting RNA (piRNA)* is the largest class of small non-coding RNA, mostly involved in the epigenetic and post-transcriptional silencing of transposable elements and other repeat-derived transcripts [85]. At least some of 28S rRNA fragments are very closely related to piRNAs, and potentially work as small guide RNA [103].

### 7.6. Double-Stranded Break-Induced RNAs (diRNAs)

Double-stranded break-induced RNAs (diRNAs) are also derived from the rDNA locus [104]. These molecules were first characterized in transgenes in Arabidopsis and cultivated human cells. The biological relevance of endogenous diRNAs remains largely unknown, but they probably take part in repair-related homologous recombination and might be important factors in the carcinogenesis [104,105].

We have not found in the literature any data about the role of the human rDNA derived gRNAs, piRNAs, and diRNAs in carcinogenesis or tumor suppression, but the above-mentioned properties of these molecules suggest such involvement.

### 7.7. Long Non-Coding RNAs (lncRNAs)

Long non-coding RNAs (lncRNAs) *produced by rDNA.* lncRNAs are RNAs over 200 nt, which may be transcribed from the coding and non-coding regions of the genome in sense and anti-sense manner [106]. Their function includes protein binding or sequestration, chromatin modifications, small nuclear/nucleolar ribonucleoprotein formation and modulation of the translation. Accumulating evidence suggests that lncRNAs are critical regulators of gene expression in development, differentiation, and human diseases, including cancer [88]. A number of such molecules are produced by rDNA locus.

### 7.8. Promoter Associated RNA (pRNA)

*Promoter associated RNA (pRNA)*, is transcribed by pol I from a region beginning at about −2 kb and stretching into the 5′ ETS [107]. The primary transcript is processed by the exosome complex to produce 150–300 nt functional molecules. pRNA is required for rDNA silencing and acts through the recruitment of the nucleolar remodeling complex (NoRC) [108]. Helicase DHX9, participating in regulation of nucleoplasmic genes, is involved in pRNA processing [109]. Additionally, a 5′ portion of pRNA may form a DNA:RNA triplex with the T0 regulatory element of the rDNA promoter. Although the triplex has been observed only in vitro, it is believed that pRNA recruits the DNA methyltransferase DNMT3b to enforce chromatin inactivation by CpG methylation [110]. Some studies indicate that even a minimal sequence changes in the pRNA producing region may relate to malignant transformation. SNV at +139 (T/C) situated within the 5′ ETS was found in epithelial cancer cells. Presence of this variant correlated negatively with the level of pRNA expression. A model using MPGAfold and Mfold programs suggested that the +139 SNV changes the secondary structure and affects the transcription. The SNV might also alter binding of glucocorticoid receptors, interact with a nearby nucleolin binding site, and participate in maintaining genomic stability [67,111].

### 7.9. Promoter and Pre-rRNA Antisense Transcript (PAPAS)

This RNA is produced in anti-sense direction by pol II from a region covering a large part of the ribosomal gene area, promoter, and enhancer. PAPAS is actually a heterogenous pool of 12–16 kb long transcripts arising from multiple start sites [112]. These RNAs are expressed as a response to the stress and inactivate pre-rRNA transcription in different ways, mainly by recruiting the histone methyltransferase Suv4-20h2 to the rDNA promoter [113]. By forming a DNA:RNA triplex, PAPAS promote nucleosome remodeling and deacetylase complex (NuRD) recruitment and H4ac inhibition. In this respect PAPAS is similar to pRNA [114]. Direct involvement of PAPAS in carcinogenesis has not been demonstrated, but its overexpression in vitro was associated with miR-34a inhibition, as well as in vitro migration and invasion of triple-negative breast cancer cells [115]. The anti-sense activity of RNA Pol II at the regions around 28 kb and around 38 kb establishes an R-loop shield, preventing RNA Pol I recruitment to the locus. Under normal conditions, the continuous transcription from these loci producing the so-called antisense intergenic ncRNAs (asincRNA) prevents the unwanted RNA pol I transcription from the same [116]. Interestingly, the non-canonical structures of another type (e.g., G-quadruplexes) also take part in the regulation of rDNA transcription activity [117].

### 7.10. Detaining lncRNAs

Three regions of IGS enriched in [TC]_n_ or [GA]_n_ simple repeats and situated at ~16 kb, ~22 kb and ~28 kb are called rIGS16, rIGS22, and rIGS28, respectively. These regions produce ~300 nt long RNAs, which in response to stress, such as acidosis or heat shock, detain within the nucleoli, and temporarily inactivate certain enzymes, including components of the DNA polymerase complex. The process is accordingly named “nucleolar detention” [118]. Since the detaining lnRNAs control DNA replication in the cell under the stress condition, their function is very likely among the first targets of malignant transformation.

### 7.11. Pyrimidine-Rich Non-Coding Transcript (PNCTR)

Pyrimidine-rich non-coding transcript (PNCTR) is another kind of lncRNA transcribed from the ~28 kb IGS region. This pol I transcript is >10 kb in length and its pyrimidine-rich sequence harbors multiple putative binding sites of polypyrimidine tract-binding protein, that is associated with primary transcripts and regulates their processing. Remarkably the PNCTR is much more abundant than the detaining lncRNAs and shows higher levels in cancer cells compared to normal cells [119].

### 7.12. Alu and AluRNA

The human genome contains about 10% Alu elements, the mutable DNA sequences of ~300 nt in length belonging to the class of retrotransposons. These elements are particularly abundant in rDNA, where they are for the most part incomplete and unable to migrate [120]. Alu repeats often contain simple repeats (3–5 bp), and their flanking sequences are hot-spots of recombination [121]. Therefore, these elements are often associated with genetic instability, especially through Alu/Alu non-allelic homologous recombination, which produces insertions, deletions, and duplications [122,123,124,125,126]. Transposable Alu elements may disrupt gene expression or produce abridged genes, and this may be a cause of malignant transformation, e.g., by damaging tumor suppressor genes [127]. In a study of relationship between Alu and the cancer it was found that recessive cancer genes (tumor suppressor genes) are enriched with Alu elements (*p* < 0.01) compared to dominant cancer genes (oncogenes) and the entire set of genes in the human genome [127]. Recently it was found that Alu elements are transcribed into Alu RNAs, which can inhibit mRNA translation [128]. On the other hand, these RNAs are abundant in nucleoli and may be required for the maintenance of the nucleolar structure [129,130]. After knockdown of Alu RNAs, production of rRNA was decreased, and the essential nucleolar proteins, such as nucleolin and pol I, were dispersed [129]. Thus, Alu RNA seems to be another factor stabilizing rDNA locus.

## 8. rDNA Regions of Unknown Function

Hypothetically, many other regions of rDNA locus may be functional even without producing ncRNAs (e.g., as specific protein binding sites or potentially transposable elements). Functional regions will be very likely found among the numerous repetitive sequences within IGS (Figure 2). These sequences include long repeats 1 and 2 (LR1 and LR2), the “butterfly” boxes, the 320 bp long sequence named thus due to a butterfly-like pattern on sequencing gels, the ActcA repeats, the 70–90 bp blocks containing CTTG and CTTT, separated by CGTG, two nameless repeats at ~28 bp and ~38 bp, as well as multiple shorter clusters of repeats (Figure 1). Some non-repetitive regions also may prove to be functional at certain conditions. Particularly interesting in this respect is the large coding DNA sequence (CDS) (Figure 2). This CDS is analogous to the cell division cycle 27 gene (CDC27) situated on the chromosome 17. Although the sequence belonging to IGS is abridged, it may be important in the study of carcinogenesis, since CDC27 seems to be involved in tumor suppression. Its transcription is down-regulated in various cancers, including breast cancer and colorectal cancer. Additionally, mir-218-2 RNA promotes glioblastomas growth, invasion and drug resistance by targeting CDC27 [131,132].

## 9. Ribosomal DNA and Cancer Therapy

It has been long known that a number of conventional chemotherapeutic agents impair ribosome biogenesis and/or nucleolar size, number and morphology. A recent screen of common chemotherapeutic drugs demonstrated that out of 36 agents tested, 21 were found to affect rDNA transcription, early rRNA processing (measured by the occurrence of the 32S rRNA intermediate) and late rRNA processing (measured by the occurrence of the mature 28S or 18S rRNAs) [27]. The platinum-containing compound cisplatin inhibits pol I transcription by cross-linking high mobility group (HMG) domains of rDNA thus preventing association of UBF with the promoter [133]. Inhibitors of topoisomerase I activity, specifically camptothecin, irinotecan, and topotecian can also efficiently disrupt pol I transcription [134,135]. Now the rDNA and its functions are becoming a specific target of the anti-cancer therapy [136]. The first of the designed drugs is CX-3543 (quarfloxin), it specifically inhibits the elongation stage of pol I transcription by preventing the stabilizing interactions between nucleolin and G-quadruplexes in the rDNA gene. Nucleolin mediates the stabilization of G-quadruplex structures facilitating rapid transcription. When this protein, selectively displaced from DNA by CX-3543 relocates from the nucleolus to the nucleoplasm, the rDNA transcription is suppressed [117]. The inhibition of the abnormally enhanced rDNA transcription can not only reduce ribosome biogenesis and protein synthesis necessary for growing cancer cells, but also restore many tumor suppressing processes, such as activation of p53 [7]. It was reported that the small molecule inhibitor of pol I transcription initiation, CX-5461, killed transformed B-cells in the lymphoma culture, while sparing the normal B-cell population [26]. CX-5461 is a G-quadruplex stabilizer and it also targets topoisomerase II beta. A Phase 1 clinical trial of CX-5461 in patients with hematologic malignancies commenced in 2013 (Peter MacCallum Cancer Centre, Melbourne). CX-5461 is also in advanced phase I clinical trial for patients with BRCA1/2 deficient tumors (Canadian trial, NCT02719977, opened May 2016). [7,137,138,139,140]. BMH-21, a DNA intercalator, which binds ribosomal DNA and inhibits transcription elongation, was suggested as another anti-cancer drug [141]. More sophisticated methods of therapy may be developed in future based on the study of the extra-ribosomal functions of the rDNA locus as a central integrator of cell proliferation and stress signaling.

## 10. Conclusions

Recent studies show that rDNA plays a crucial role in carcinogenesis. The locus has been described as a target for the essential pathways of malignant transformation. In this respect, the data on the nucleolar proteins, gain/loss of the copy numbers, abnormal transcription enhancement, and promoter methylation/demethylation seem to be conclusive. Moreover, it is believed that alterations in the rDNA structure and functions may be actual drivers of the malignant phenotype [7,26]. Some authors indicate the possibility of finding even the primary causes of cancers in the instability of the genetic or epigenetic status of rDNA [142]. But this hypothesis still has not been proved. Decisive results may come from the study of the functional regions related to both ribosomal and extra-ribosomal functions of the locus. Emerging oncological significance of these regions suggests that they require much more attention. At present, the studies are incomplete and disconnected. It is supposed that rDNA actually produces the whole classes of unknown non-coding RNAs [104]. But the already accumulated data on rDNA suggest potential practical significance of its study. Mutations in the ribosomal genes and IGS can be used in diagnosis and prognosis of the tumors. Clinical tests of the drugs inhibiting pol I transcription have already begun [7]. Perhaps the curative value of such drugs will remain limited, since they are toxic to all metabolically active cells. However, more subtle methods of treatment may be developed based on a comprehensive study of rDNA locus as a whole and, in particular, its functional regions.

## Figures and Tables

**Figure 1 cells-10-03452-f001:**
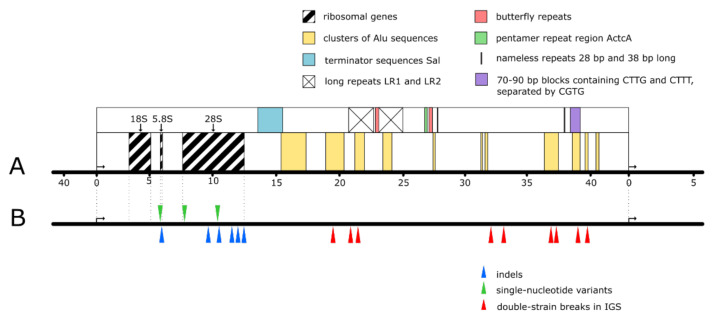
Fragile sites of human rDNA locus. (**A**) Ribosomal genes, repetitive sequences, clusters of complete and abridged Alu sequences. (**B**) The main hot-spots of double-strain breaks in IGS (red wedges) and of SNVs (green wedges) and indels (blue wedges) in the ribosomal part (the shown positions of the sites are provisional, since the sequencing of human rDNA locus is not completed).

**Figure 2 cells-10-03452-f002:**
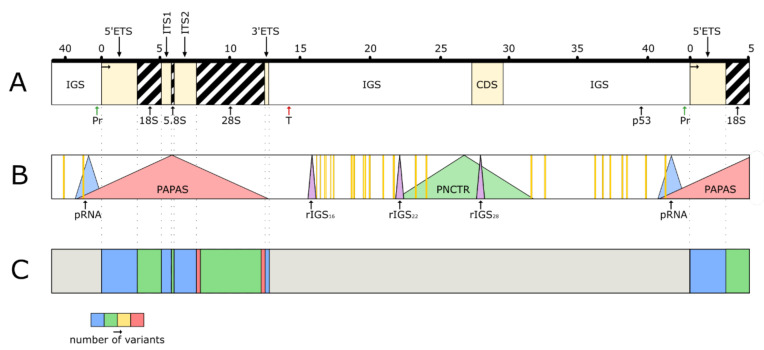
Functional regions of the human rDNA locus. These regions cover practically the entire unit and overlap each other. (**A**) A part of the cluster containing one whole unit. Pr-promoter, T-terminator, CDS-coding DNA sequence analogous to cell division cycle 27 (CDC27) gene, p53-consensus binding site for p53 protein. (**B**) Regions coding for the long non-coding RNAs (lncRNAs). Vertical orange bars represent sequences which may produce AluRNAs. (**C**) Regions coding for the small ribosomal RNAs (srRNAs). The thermogram shows relative density of srRNAs in various parts of the unit. The grey color in IGS refers to the great variability of expression and insufficient study of the srRNA sequences (the shown positions of the sites are provisional, since the sequencing of human rDNA locus is not completed).

## Data Availability

No new data were created or analyzed in this study. Data sharing is not applicable to this article.

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
