# Peer review of "Human rDNA and Cancer"

_cells, 2021, doi:10.3390/cells10123452_

Round 1

Reviewer 1 Report

The review by Smirnov et al. focus on the multicopy genes encoding ribosomal RNA (rRNA genes or in short rDNA) and their link to human cancers. The authors deal specifically with several topics of rDNA and cancer biology, including oncoproteins and tumor suppressors interacting with rDNA, rDNA methylation and instability, as well as non-rRNA transcripts originating from the locus. All of these aspects are interesting and relevant for our understanding of how rDNA and its dedicated organelle, the nucleolus, are involved in malignant transformation of cells. In general, the manuscript is informative, but remains often superficial by listing several related studies rather than connecting their conclusions. Moreover, in some parts more recent literature should be cited and discussed. My suggestions to improve the manuscript are as follows:

Abstract/Introduction:

  • Please clarify the nomenclature for ribosomal RNA, rRNA genes and rDNA. The term rDNA and rRNA are not introduced properly. It is also confusing to name the sequences encoding 18S, 5.8S, and 28S rRNAs as individual genes (which implies own regulatory sequences like gene promoters and so on).
  • Please clarify the statement: “… the pol I begins to transcribe ribosomal genes, making primary nucleoli…” (line 36). Formation of nucleoli is a consequence, but not a direct product of pol I transcription.
  • Please use another phrase for “may be blamed” in line 46.

Part 2 Nucleoli and cancer:

  • Use the English word for “organella” (line 52).
  • What is meant by “through coordinated transport of p53 and ribosomal subunit” in line 70?

Part 3 Proteins associated with rDNA and cancer

  • Please emphasize the protein stated at the beginning of each paragraph by writing in italics.
  • Could you please clarify the statement about p53 in lines 114-116: “Mutated forms of this protein appear in approximately half of all human tumors, and even when the wild type is still preserved, its function is compromised in most cases”. I guess you mean that mutation of one TP53 allele leads to functional disruption of most of the tetrameric complexes in the cell.
  • In line 131, please correct “basonectin” and remove “3.1.”.

Part 4 Methylation of rDNA and cancer

  • This part would benefit from a short general introduction of the role of rDNA (promoter) methylation. How is it established and distributed among the several rDNA repeats?
  • Please correct: “subtler changes” (line 159)
  • The sentence “Thus, methylation of rDNA locus, especially its promoter, is one of the key factors of carcinogenesis, but the phenomenon is complicated and not sufficiently understood.” in lines 164 and 165 sound overstated.

Part 5 Causes of rDNA instability

  • Regarding the “Pleiades”, I would replace “(unlike the celestial ones)” with “(like the brightest celestial ones)” in line 173.
  • Line 189, “genes for methyltransferases”: Please be more specific.
  • Please clarify: “in the region between −388 bp and +306 bp, which includes sequences involved in the inhibition of the rRNA synthesis (lines 196-198).” It would be good to mention that the positions refer to the transcription start site (I guess?). Why is the region around the transcription start site involved in inhibition of rRNA synthesis?

Part 7 Functional regions of rDNA and cancer (Fig. 2)

  • Please remove the citation of Fig. 2 from the title of this part. The title is also not very fitting, as non-rRNA transcripts derived from rDNA are discussed. (Encoding rRNA is actually the main function of rDNA).
  • Line 290: Please correct “myelodysplastic syndrome” to “5q-myelodysplastic syndrome”.
  • Line 310-315: The part about “gRNA” sounds cryptic; I would leave it out.
  • Line 335 and 336: “Promoter associated RNA (pRNA), is transcribed by pol I from a region beginning at about −2 kb and stretching into the 5’ ETS.” In humans, the transcription start site of pRNA is NOT about 2 kb upstream of the pre-rRNA start site! With regard to processing of pRNA, a recent study showing the involvement of DHX9 should be mentioned (PMID: 28588071)
  • The part about “detaining lncRNAs” (lines 367-374) is cryptic. These IGS transcripts detain proteins in the nucleolus, but do not get detained themselves. They are also not upregulated upon “hypotonic stress” and “inactivate certain enzymes”. How do these transcripts “control DNA replication”?

Part 8 rDNA regions of unknown function

  • This part is not very information. Moreover, it is closely related to Fig. 1 and should be positioned in the manuscript accordingly. Please give more background about the “butterfly boxes”. Is the CDC27 CDS transcribed in a functional mRNA?

Part 9 Ribosomal DNA and cancer therapy

  • Inhibition of pol I transcription as an anti-cancer strategy has come into focus for some time now. Thus, this part of the manuscript does not add much new insights. At least, another relative new inhibitor, BMH-21, should be mentioned. Moreover, new research on CX-5461, showing its impact on G-quadruplexes and topoisomerase II beta should be discussed.

Part 10 Conclusions

  • Lines 459-461: “It is supposed that rDNA actually produces the whole classes of unknown non-coding RNAs”. What classes of RNA are meant?

General:

  • The term “extra-ribosomal functions” of rDNA is heavily used throughout the text. However, these functions are nowhere explained in more detail.

Author Response

Dear reviewer, 
thank you for your comments and suggestions. See our reply in attached document.

Evgeny Smirnov

Reviewer 2 Report

Smirnov and coathors offer us an extensive review of the relationship between ribosomal RNA genes and cancer. The paper is correctly written and although sometimes it gets too technical, it is still readable by a less expert audience. I have a few suggestions that the authors may consider:

lines 66-67: you mention extra-ribosomal functions, but, could you describe some of these?

lines 93-94: how do chemotherapy drugs affect ribosome biogenesis?

line 162: which alternative ways can alter rDNA transcription? Could you cite some?

lines 174-175: could you explain briefly what do the CTCF protein and the H34me3 mark? In which processes are they involved?

line 180: this is one of the few mentions to extra-chromosomal circles, and rRNA genes are well-known eccDNA producers. Consider extending on this topic a bit more

lines 189-190: can you give an example of such environmental factors?

lines 199-200: can you give specific examples in which rDNA instability is linked to cancer or severe disease?

line 204: where do these "multiple changes" in rDNA sequence occur? Can you be more specific?

lines 223-226: Can you provide a numerical range of variation of rDNA copy number in normal and in pathological  conditions. I would suggest to use "in" instead of "to" in the sentence "One may reasonably expect that changes to rDNA copy number..."

line 444: I found a more recent citation of the use of this drug in https://www.nature.com/articles/s41467-020-16393-4 and https://www.nature.com/articles/ncomms14432, could you comment on these too?

line 462-463: could you give/remind us on an example of the use of rRNA genes and IGS as diagnostic or prognostic of tumours?

Author Response

(The authors gave the same response as above.)
